# Printable homocomposite hydrogels with synergistically reinforced molecular-colloidal networks

Austin H. Williams[1,4], Sangchul Roh [1,4], Alan R. Jacob [1], Simeon D. Stoyanov [1,2,3], Lilian Hsiao [1] & Orlin D. Velev [1✉]

The design of hydrogels where multiple interpenetrating networks enable enhanced mechanical properties can broaden their field of application in biomedical materials, 3D printing, and soft robotics. We report a class of self-reinforced homocomposite hydrogels (HHGs) comprised of interpenetrating networks of multiscale hierarchy. A molecular alginate gel is reinforced by a colloidal network of hierarchically branched alginate soft dendritic colloids (SDCs). The reinforcement of the molecular gel with the nanofibrillar SDC network of the same biopolymer results in a remarkable increase of the HHG's mechanical properties. The viscoelastic HHGs show >3× larger storage modulus and >4× larger Young's modulus than either constitutive network at the same concentration. Such synergistically enforced colloidal-molecular HHGs open up numerous opportunities for formulation of biocompatible gels with robust structure-property relationships. Balance of the ratio of their precursors facilitates precise control of the yield stress and rate of self-reinforcement, enabling efficient extrusion 3D printing of HHGs.

[1] Department of Chemical and Biomolecular Engineering, North Carolina State University, Raleigh, NC, USA. [2] Physical Chemistry and Soft Matter, Wageningen University, Wageningen, The Netherlands. [3] Department of Mechanical Engineering, University College London, Torrington Place, London, UK. [4] These authors contributed equally: Austin H. Williams, Sangchul Roh. ✉email: odvelev@ncsu.edu

Hydrogels made of polymer networks in water are generally soft and brittle, often lacking the resilience and toughness required for widespread deployment in applications such as tissue scaffolds, food products, soft robotics, and flexible electronics[1–10]. The elasticity of these materials can be improved by combining interpenetrating covalent and ionic polymer networks to form highly stretchable and tough hydrogels[11–17]. Another way of improving and enhancing the mechanical properties of hydrogels is by including high aspect ratio fillers, such as fibers, to mechanically reinforce the gel matrix[18–26]. Fiber-reinforced hydrogels can serve as comprehensive bioscaffolds for cell growth. The fiber networks offer robust control of hydrogel stiffness, and can mimic the morphology of the physiological extracellular matrix, which has been shown to affect cell activity, differentiation, and response[27,28]. Controlling hydrogel stress propagation by fiber reinforcement is also useful in applications such as soft robotics, as it may enable nonlinear deformation modes and durability against frictional sliding[29]. However, using fillers made of a different material from the matrix introduces stressed interfaces that cause crack formation when the composite hydrogels are strained or heated.

One different materials design approach is to use single-polymer composites, or so-called homocomposites. The mesoscale reinforcement network of homocomposites is made of a material that is chemically identical to that of the primary matrix. Homocomposite reinforcement networks allow modulation of the mechanical properties of the host matrix without stress concentrations, delamination points, or other problems arising from interfacial incompatibilities[30,31]. Homocomposite formulations have also been designed using elastomer particles bound by capillary bridges of the precursor liquid elastomer in multiphasic silicone pastes for 3D printing[32,33]. The chemical similarity of the continuous and reinforcement phases provides superior adhesion and enhanced load transfer[25,34–36]. Previous studies have also shown that homocomposites with fiber reinforcements allow tailored bulk melt rheology along with increased tensile modulus and strength[37,38]. They are made by using the differences in melting temperatures between thermoplastic polymer crystalline phases as a processing window[39,40]. However, homocomposite hydrogels (HHGs) remain challenging to fabricate because of the lack of techniques to construct reinforcing networks with the same chemical composition as the hydrogel matrix.

We report a class of HHGs where both the primary gel matrix and the reinforcement network are made of sodium alginate (SA). Alginate hydrogels are used in a broad range of technological applications such as biomedical implants, tissue engineering scaffolds, and food products[7,41]. These HHGs are reinforced by a fibrillar network of alginate soft dendritic colloids (SDCs). The SDCs are a hierarchically structured class of soft matter synthesized through a scalable process of shear-driven precipitation in a turbulent medium, described in our recent report[42]. The high degree of branching around the cores of the SDC particulates makes them morphologically similar to polymeric molecular dendrimers, but SDCs are orders of magnitude larger than these dendrimers[43,44]. Our previous studies with polymer SDCs dispersed in an organic oil medium showed that the branched, nanofibrous structure of the dendricolloids enables efficient network formation leading to gelation at low volume fraction due to enhanced van der Waals interactions[42,45].

The hierarchically branched SDCs are expected to serve as efficient reinforcement in composite materials. Their branches provide large surface area, which could increase the composite stability by distributing the stress more uniformly[42,46–50]. We investigate here the SDC's effectiveness in reinforcing homocomposite hydrogels, expecting that the high surface area and maximized adhesiveness between the SDC network and the matrix will result in hydrogels of outstanding mechanical properties. To analyze the unusual mechanical properties of the HHGs, we also perform shear and tensile tests on their individual constituents: aqueous SA SDC suspensions and crosslinked SA molecular hydrogels (CMHs). We then characterize HHGs composed of SA SDCs embedded in a SA CMH matrix and discuss the origins of their synergistically enhanced viscoelastic properties. On this basis, we demonstrate that the HHGs can be readily formulated to provide a yield stress for 3D printing or pressure-driven extrusion.

## Results

**Formation of soft dendritic colloids from alginate**. The alginate SDCs used as reinforcement networks were fabricated using a turbulent shear-driven precipitation process for the manufacturing of polymer dendricolloids[42]. In order to obtain hydrogel SDCs, a solution of alginate (120–190 kDa) was injected in an aqueous solution of $Ca^{2+}$ ions, which effectively bind and crosslink two -COO- side groups on the alginate backbone (Fig. 1a). The turbulent shear precipitation process results in SDCs with characteristic hierarchical morphology, illustrated in Fig. 1b, with multiscale branching and generations of fibers spanning roughly 3 orders of magnitude in length scale. SDCs consist of micron-scale fibers that branch multiple times into ever thinner fibers. The outermost layer surrounding each SDC, or so-called "corona," encompasses flexible nanofibers that can be as thin as 10 nm (Fig. 1b). The nanofibers in the coronae endow them with physical adhesiveness, which is a major factor in the dendricolloids' ability to build the structural strength of the colloidal network. The effective overall size of common SDCs, including their coronae, is in the 100–500 μm range. The macroscopic features and properties of the SDCs and their suspensions are consistent across bulk measurements (for more details see Supplementary Figs. 1 and 2 and corresponding SI text). The SA SDCs suspensions were washed with deionized water to remove excess $CaCl_2$ ions from the suspension.

**Viscoelastic properties of aqueous SDC suspensions**. We first established whether alginate SDCs form colloidal network hydrogels at low volume fractions in water. The hypothesis is that in aqueous suspensions, the SDCs in contact will adhere strongly through van der Waals forces to form a percolating network of branched fiber sub-contacts (Fig. 2a)[42]. Characterization of the storage ($G'$) and loss ($G''$) moduli of SDC suspensions within the linear viscoelastic region using a stress-controlled rheometer showed that the SDCs have a strong propensity for forming colloidal networks. A yield stress was observed in aqueous suspensions of 0.25 wt.% SDCs, i.e., at a lower concentration than most types of conventional colloidal gels (Supplementary Fig. 2)[51,52]. The slopes of $G'$ and $G''$ obtained from small amplitude oscillatory frequency sweeps showed that SDC suspensions with solid content above 0.4 wt.% are gel-like in nature (Fig. 2b)[53–55].

The efficient networking of SDCs is likely a result of the abundant adhesive physical contacts between the flexible, branched dendricolloids[42], a phenomenon known as contact splitting. It is one of the reasons for the universal adhesiveness of fibrillar structures such as the nanofiber-padded footpads of the gecko lizards[56]. SDC suspensions exhibit more pronounced solid-like characteristics than suspensions of common alginate particles. A 1.0 wt.% SDC suspension has a value of $G' = 200$ Pa as compared to reported values of $G' = 10 – 100$ Pa for 1.0 wt.% alginate microgel suspensions[57–59].

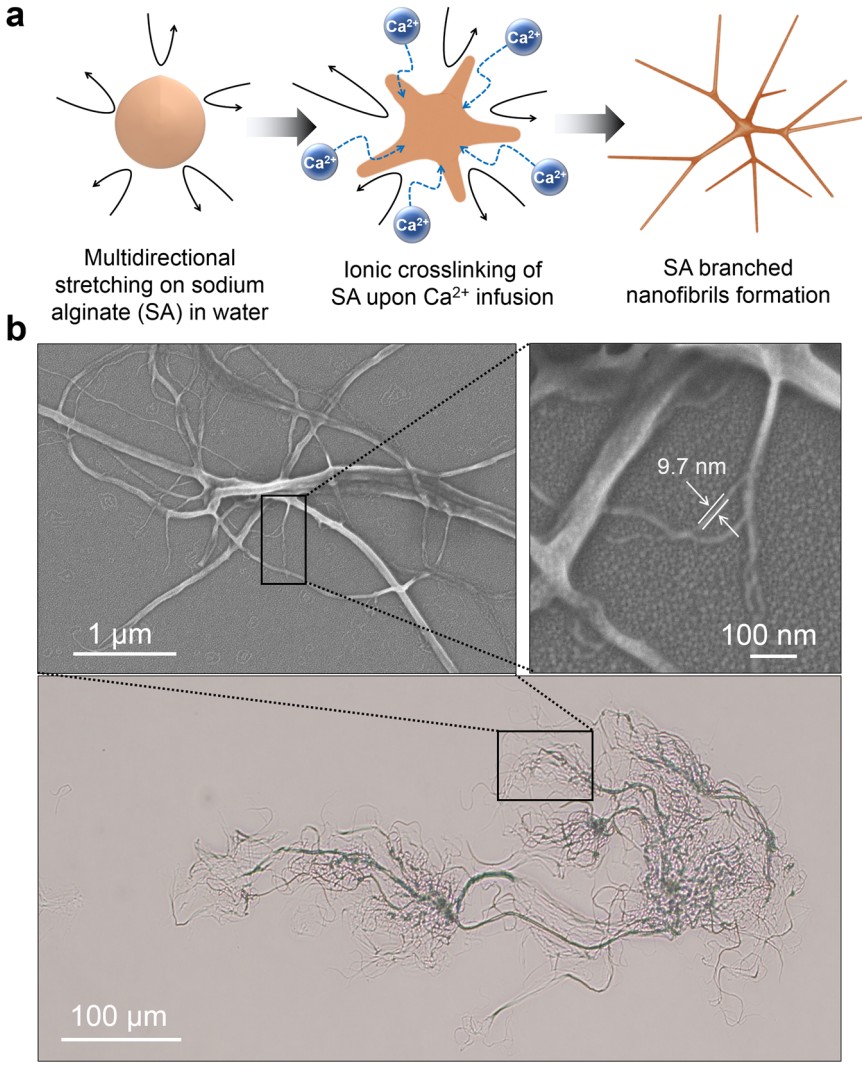

**Fig. 1 Turbulent shear-driven precipitation and morphology of a sodium alginate soft dendritic colloid. a** Schematic of the SA SDC formation process by precipitation in a turbulent medium. **b** Optical microscopy and field emission scanning electron microscopy (FE-SEM) images of sodium alginate soft dendritic colloids. These images illustrate the hierarchical fibrillar morphology of the SDCs, which leads to high exclusion volume, strong van der Waals adhesiveness, and high propensity for gelation.

**Viscoelastic properties of alginate crosslinked molecular hydrogels (CMHs).** The continuous phase of the HHGs consists of a molecular alginate gel crosslinked with $Ca^{2+}$ ions. We first characterized the properties of molecular hydrogels made of 1.0 wt.% crosslinked alginate in the absence of SDCs. The CMHs were prepared by the addition of $CaCO_3$ nanoparticles and D-Gluconic acid δ-lactone (GDL) to a SA solution. As GDL undergoes hydrolysis and lowers the pH, the $CaCO_3$ slowly releases $Ca^{2+}$ ions[60,61]. The linear viscoelastic properties of the CMHs following 2 h of equilibration are plotted in Fig. 2c. Solid-like behavior is observed above 0.05 wt.% $CaCO_3$ (20/1 SA/$CaCO_3$). Further loading of the alginate hydrogel with $Ca^{2+}$ increases its stiffness; however, we found that hydrogels containing ≥0.2 wt.% $CaCO_3$ (5/1 SA/$CaCO_3$) exhibited significant syneresis—pronounced shrinkage of the crosslinked gel accompanied by the release of water (Supplementary Fig. 3). To avoid such syneresis, we maintained the ratio of 10/1 SA/$CaCO_3$ (1.0 wt.% SA, 0.1 wt.% $CaCO_3$ for pure CMH) in the HHG formulations. Interestingly, at this ratio the pure 1.0 wt.% CMH and 1.0 wt.% SDC gels had storage moduli of similar magnitude, even though these two systems are gelled by different physical mechanisms and are very dissimilar in their appearance (clear

CMH gels vs. turbid SDCs suspensions) attributed to the $Ca^{2+}$ being highly localized in the SDC fibers as opposed to homogenously distributed in the CMH matrix.

**Viscoelastic properties of composite HHGs.** We next investigated how the colloidal network of SA SDCs can be combined with the molecular SA CMH matrix and crosslinked to produce self-reinforced HHGs. We synthesized a variety of HHGs where the total SA concentration was kept constant at 1 wt.% while the ratio of SDCs to CMH was varied (Supplementary Table 1). The $G'$ and $G''$ values of the HHGs were measured as a function of time using small amplitude oscillatory stress and frequency sweeps after an equilibration time of $t = 2$ h. All HHG samples exhibited solid-like behavior (Fig. 3a). Notably, the $G'$ values of all HHG samples were larger than those of either pure CMHs or pure SDC suspensions, with maximum values recorded at low SDC concentrations (0.175–0.25 wt.% SDCs). Representative stress–strain curves (Fig. 3b) obtained by mechanical tensile testing also demonstrate that the HHG systems have larger stiffness than the gels from SDCs or CMHs alone. Interestingly, we were not able to measure reliably the tensile behavior of pure SDCs suspension using a common testing machine. The SDC

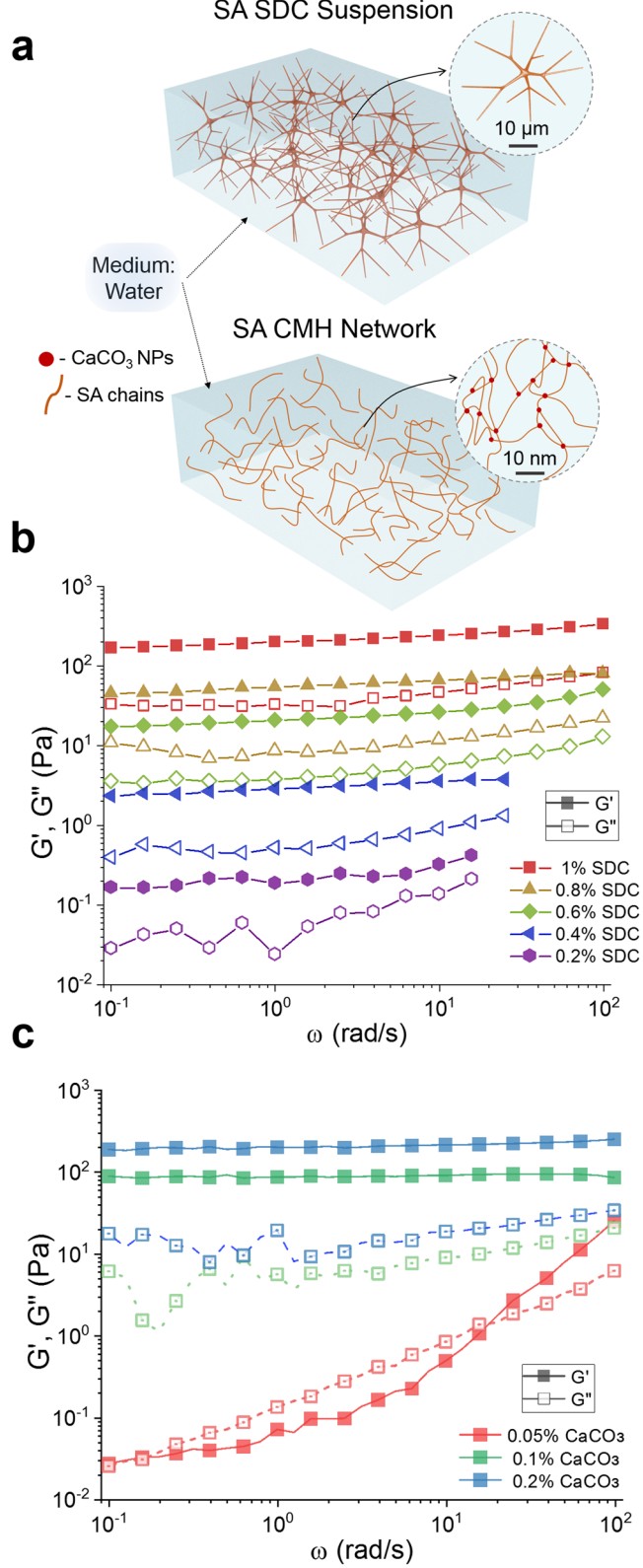

**Fig. 2 Linear viscoelasticity and yield stress of SA SDC suspensions and molecular gels. a** Scheme comparing the gelation mode of a viscoelastic suspension of SDCs and of a crosslinked SA molecular hydrogel (CMH). Note the different length scales of the networks. **b** Frequency sweeps indicating the storage modulus ($G'$, filled) and loss modulus ($G''$, unfilled) of SA SDCs in aqueous suspensions. **c** Frequency sweeps indicating the storage modulus ($G'$, filled) and loss modulus ($G''$, unfilled) of 1.0 wt.% SA CMHs with 0.05 wt.%, 0.1 wt.%, and 0.2 wt.% $CaCO_3$ nanoparticles and 28.14 mM GDL after equilibration for 2 h.

the values of the complex modulus $G*$, and the Young's modulus, $E$, for the homocomposite gels showed greater than 3× increases with maxima at low SDCs to CMH ratios. This effect cannot be solely attributed to the slightly increased concentration of the $Ca^{2+}$ crosslinker in the homocomposite system with $Ca^{2+}$ localized both within the SDC reinforcement as well as homogenously distributed through the CMH continuous phase. The highest shear modulus measured in HHGs ($G* = 950$ Pa at a composition of 0.125 wt.% SDCs/0.875 wt.% CMH) does not correspond to the highest $Ca^{2+}$ concentration, as further increasing the SDC content reduces the resulting HHG stiffness. The $Ca^{2+}$ concentration in this formulation was estimated to be ~1.5× the concentration found in the pure CMH (Supplementary Fig. 4, Supplementary Table 2). A frequency sweep performed on a CMH with 2× $CaCO_3$ shows that the $G* = 250$ Pa (Fig. 2c), which is much less than that of the stiffest HHG formulation (0.125 wt.% SDCs/0.875 wt.% CMH). Multiple measurements consistently showed a small increase in the $G*$ peak at a composition of 0.875% SDCs, which could be a result of secondary entanglement of the concentrated SDCs stabilized by small amounts of molecular alginate, although its exact mechanism has yet to be elucidated.

In summary, the strong synergistic effect leading to increased mechanical strength of the HHGs could be directly attributed to the physical entanglement of the molecular SA and colloidal SDC networks, where the divalent $Ca^{2+}$ ions bridge the alginate chains in the egg-box configuration at the interface between the microgel fibers and the continuous matrix[59]. The resulting HHG structure is illustrated in Fig. 3d. This divalent-ion crosslinking structure is stable in most media but can be disassembled by immersion in solutions of strong chelating agents such as EDTA. We found that both pure SA and SDC systems dissolve easily in EDTA, while the HHG is more resilient, possibly due to suppressed chelator transport (Supplementary Fig. 5).

**Time-dependent stiffening of homocomposite hydrogels**. One of the other functionalities imparted by the colloidal-molecular networks in the HHGs is in the tunability of their gelation kinetics. We measured the time dependence of gelation through oscillatory shear rheology performed on SDCs, CMHs, and composite HHGs at the same SA concentration of 1 wt.%. The time sweeps performed with a pure 1 wt.% SDC suspension plotted in Fig. 4a show that the SDC suspension immediately exhibits solid-like behavior without the addition of $CaCO_3$ or GDL. This is expected, as the formation of this network occurs by contact splitting and entanglement of the fibrillar dendricolloids. The values of $G'$ and $G''$ of this system do not show any further stiffening by crosslinking or densification. The pure CMH, on the other hand, initially shows liquid-like behavior and gradually solidifies as $Ca^{2+}$ crosslinker is released by hydrolysis. It develops into a fully crosslinked system in ~120 min (Fig. 4b).

The temporal evolution of the HHG is directly linked to the kinetics at which its constituent SDCs and CMH assemble into networks. The HHG initially solidifies due to the gelation of the

particulate gel is brittle and lacks the intrinsic elasticity of the crosslinked molecular alginate gel, illustrating the difference in their physical networking mechanisms.

The data from both rheometry and tensile stress–strain measurements for all HHGs are consolidated in Fig. 3c. These data demonstrate that the homocomposite systems comprising mixed SDCs and CMHs exhibit a strong synergistic effect. Both

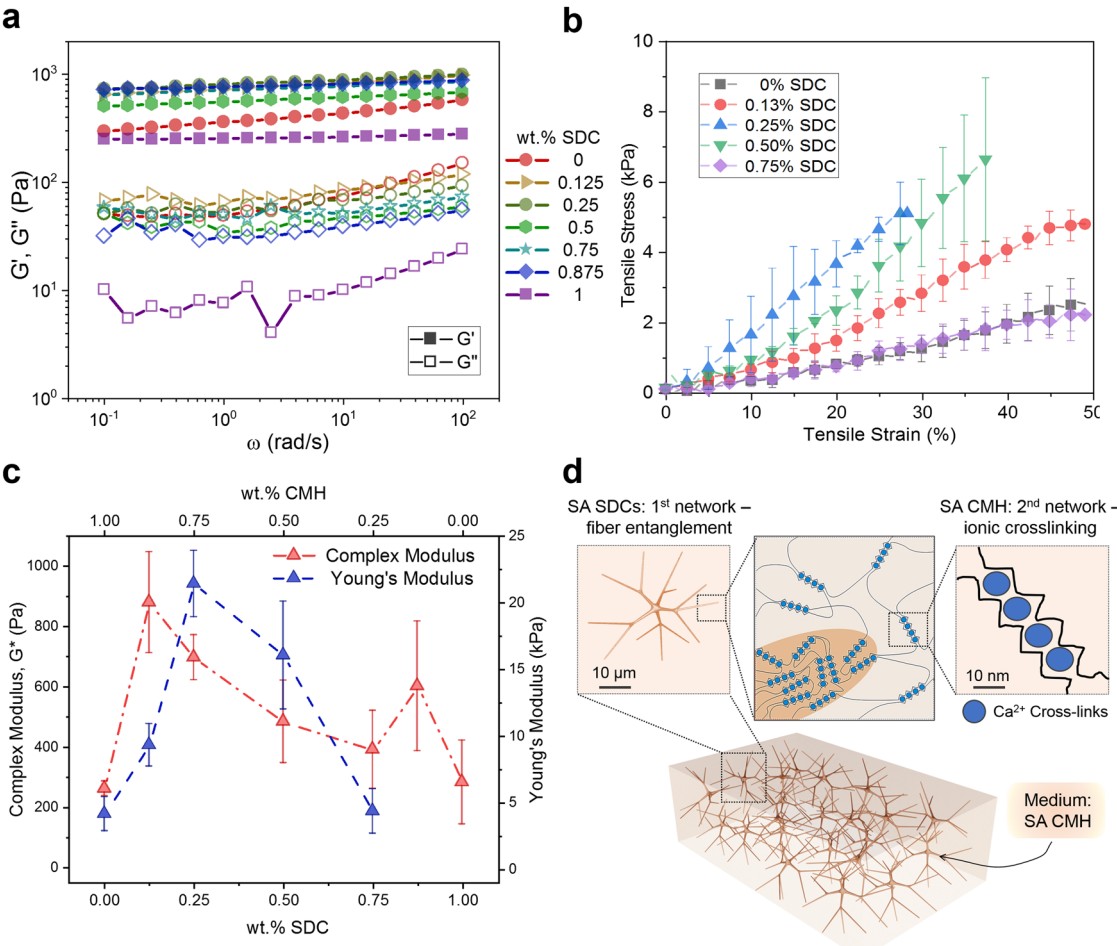

**Fig. 3 Mechanical properties of SA homocomposite gels. a** Frequency sweeps of 1.0 wt.% HHGs of varying SDC concentrations (typical curves with data consistency within 10%). **b** Representative stress–strain curves for various HHGs ($n = 5$). **c** Rheological complex modulus obtained from a frequency sweeps at a frequency of 1 rad/s (red, $n = 3$) and tensile (blue, $n = 5$) properties of various SDC/CMH HHGs. Total alginate concentration is maintained at 1.0 wt.% for all data. **d** Schematic of the hierarchical interactions between the colloidal and molecular alginate networks in HHGs. These results demonstrate the structural synergism of the double SDC and CMH networks within HHGs resulting in controllable and self-reinforced high composite stiffness.

mechanically rigid SDC network. It then forms a sturdier hydrogel over time as the interpenetrating CMH molecular network becomes crosslinked by $Ca^{2+}$ ions released in situ (Fig. 4c, d). The final value of $G'$ of the HHG system is ~3× larger than that of the pure SDCs, in agreement with the mechanical data reported in Fig. 3. These material properties reveal controlled initial yield stresses and slow buildup of hydrogel elasticity over time. This makes them particularly suitable as 3D printable materials for making of intricate and sturdy hydrogel architectures.

**3D printing with HHGs**. The homocomposite approach allows precise control over the properties of the mixture in a way that enables 3D printing via extrusion, which has traditionally been challenging with hydrogel precursors. Notably, neither of the two components of the composite gel are viable as printing inks on their own. Pure SDC suspensions are mechanically stable upon extrusion and maintain their shape upon removal of an applied stress. However, suspensions made of SDC particles alone cannot be crosslinked by GDL-CaCO₃ addition to produce a stiff elastic material by bridging active -COOH- sites on neighboring SDCs (Supplementary Fig. 6). Conversely, non-crosslinked CMH solutions are unsuitable for extrusion additive printing because

they slowly form solid gels by GDL-CaCO₃ crosslinking (Fig. 4b), a solidification process that requires time on the order of hours to days. By making a homocomposite ink using these two distinct forms of alginate, we present a type of bespoke, extrudable hydrogel material in which the time-independent yield stress and solidification time can be tuned on demand. As the 3D printer applies a pressure drop that is greater than the yield stress of the HHGs, the extruded shape is preserved by the rapid gelation of the SDC network (Fig. 4c, Supplementary Movie 1).

To demonstrate the extrudability and shapeability of such hydrogels combining colloidal and molecular networks, HHG formulations with total alginate concentration of 1.5 wt.% were loaded into a 3D printer operating by pressure-driven extrusion (Fig. 5a). The homocomposite systems were amenable to 3D printing in ambient conditions without necessitating a crosslinker bath or yield stress medium to maintain its shape. Notably, the $G'$ of 1500 Pa of the pure SDC suspension is nearly four orders of magnitude larger than that of the CMH mixture at the same concentration prior to GDL addition (0.5 Pa, Fig. 5b).

The mixing of SDCs and CMHs results in the emergence of yield stress at all ratios tested. At small strains, the network remains a solid-like material due to the van der Waals interactions. As the strain increases beyond the yield point, the physical entanglements between the SDC fibrils are overcome and

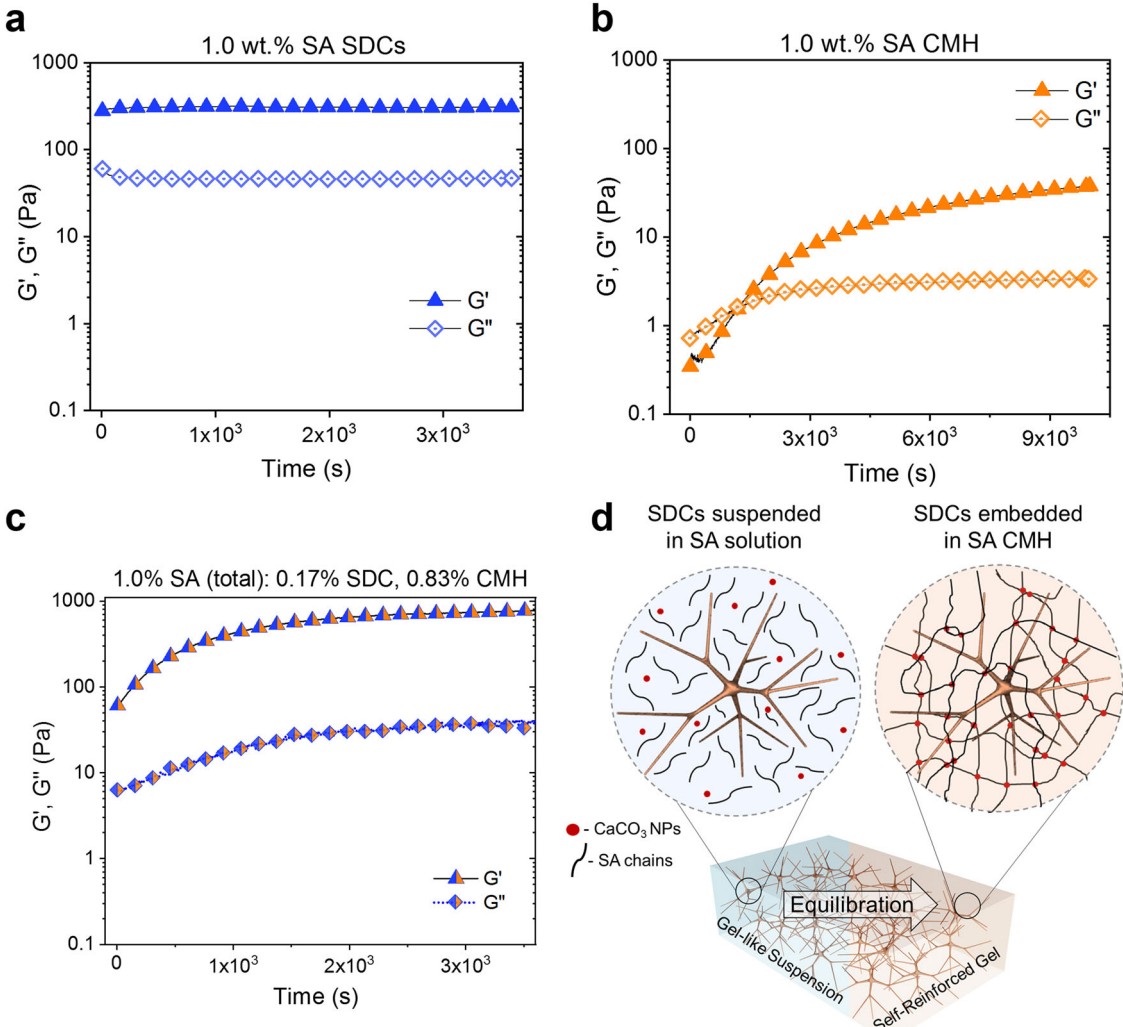

**Fig. 4 Time-dependent gelation of SA SDC suspensions, SA CMHs, and SA HHGs.** Time sweeps indicating $G'$ and $G''$ (1 rad/s, 1% strain) of **a** 1.0 wt.% SA SDC suspension in water, **b** 1.0 wt.% SA CMH following GDL addition and **c** 1.0 wt.% total SA (0.17 wt.% SDC, 0.83 wt.% CMH) HHG following GDL addition. **d** Schematic showing the time-dependent transition of a HHG from a colloidal gel to a self-reinforced homocomposite via CMH crosslinking. HHG formulations show immediate, elastic-solid behavior from the colloidal network and further stiffen with the ionic crosslinking of the molecular network.

the yielded material can be extruded. Despite maximal HHG gel stiffness emerging at lower relative SDC/CMH ratios (Fig. 4c), HHG formulations with greater relative SDC contents produced more pristine filaments with improved filament layering (Supplementary Movie 2). The maximum $G^*$ obtained at a 1/1 SDC/CMH ratio is consistent with the results of equilibrated 1.0 wt.% HHGs, which show a similar maximum $G^*$ at that ratio (Supplementary Fig. 7). The increased yield stress of 1.5 wt.% SA HHGs enabled smooth and consistent HHGs extrusion for 3D printing.

The process of 3D printing of a homocomposite hydrogel by direct extrusion in air is demonstrated in Fig. 5a, c. The HHG formulation is extruded through a nozzle (25 G, 0.26 mm ID) at 140 kPa and the gel retains neatly its shape due to its yield stress (≈80 Pa). Additive shaping of the structures in the z-direction can be achieved by overlaying consecutive layers, which were found to adhere well to the underlying ones. We were able to achieve the additive printing of >10 layers of hydrogel in the vertical direction without reducing the rate of extrusion. A few examples of such vertically developed structures extruded in consecutive layers are demonstrated in Fig. 5c. Much higher height/width ratios can be achieved by staged extrusion, which allows additional time for gel

solidification, as well as increasing the paste yield stress via the HHG composition. Following molecular crosslinking and stiffening for 60 min, the printed hydrogel shapes could be readily removed from the substrate (Fig. 5d). Further adjustment of the relative concentrations of SA, SDCs, $CaCO_3$, and GDL allows the formulation of printable homocomposite hydrogels with tunable properties.

## Discussion

We introduce a class of homocomposite hydrogels that are self-reinforced by a network of alginate SDCs interspersed within an alginate CMH network. The dispersed and adhesive nanofibrillar coronas of the SDCs lead to the formation of strong gels in SDC suspensions at very low volume fractions. They enable the formation of sturdy HHGs when interspersed in a continuous, molecular gel of the same biopolymer. The stiffness of the resulting HHG appears to be governed by the synergistic interactions of the molecular alginate network and the colloidal-scale fibrillar SDC network. These HHGs are made using SA as a common polymer, but the principle can be applied to other hydrogels of natural or synthetic origin. While methods for 3D

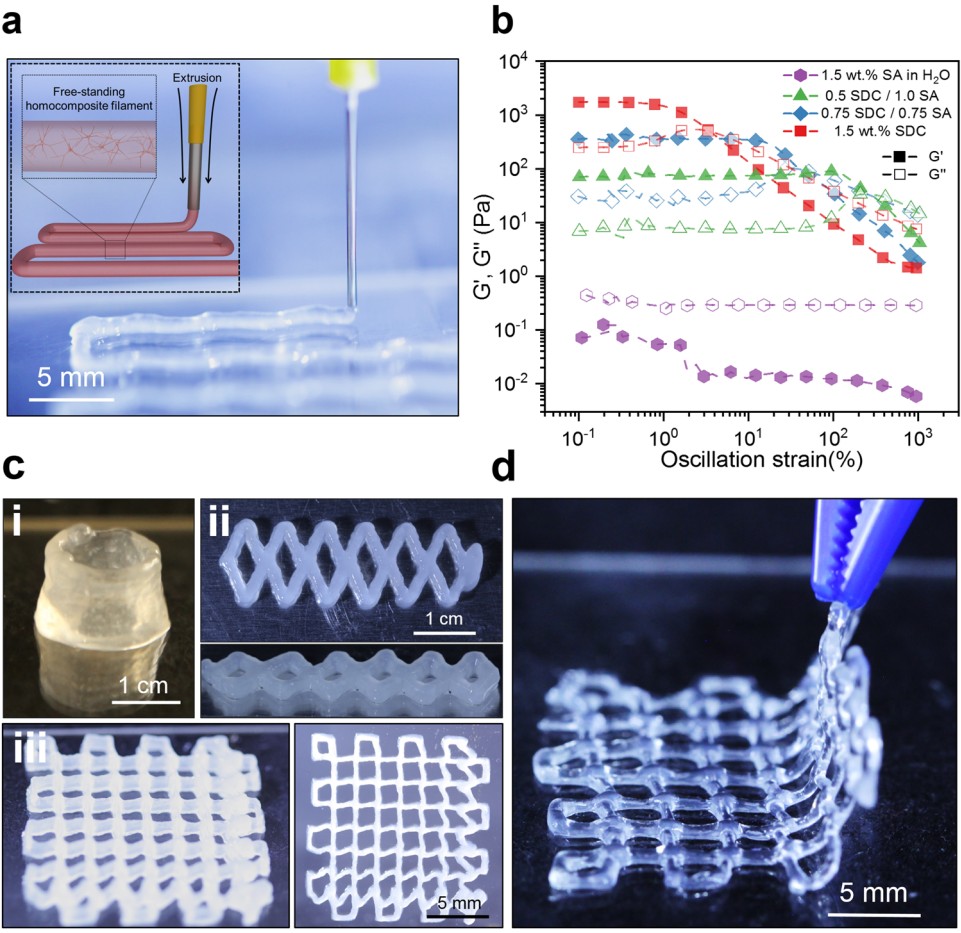

**Fig. 5 3D printing with homocomposite hydrogels. a** Schematic illustration and image of 3D printing of SA HHG ink. **b** Complex modulus vs. stress curves for 1.5 total wt.% SA and SDC/SA HHG mixtures containing $CaCO_3$ nanoparticles, prior to GDL addition measured at 6.23 rad/s. **c** Images of extruded HHG paste that has been layered in 3D (0.75 wt.% SDC, 0.75 wt.% CMH): (i) 10 layers from a 18 G nozzle, (ii) 3 layers from a 16 G nozzle, (iii) 5 layers from a 25 G nozzle. **d** Removal of the equilibrated hydrogel design from the substrate. HHG formulations have the ability to flow upon extrusion, maintain their shape due to the colloidal yield stress, and result in a strongly self-reinforced structure following completion of the secondary ionic crosslinking reaction.

printing with alginate have been reported previously[41,62], the method presented here has a number of potential advantages, such as it does not need any bath of crosslinking medium, as well as printing with pure alginate without any additives. The HHGs in this study demonstrate two distinctive emerging concepts that can guide materials design of hydrogel compositions with precisely adjustable properties. Firstly, the HHGs are stabilized by interpenetrating double networks at two very different length scales: molecular (nanometer scale) and colloidal (up to microscale). Second, due to the chemical uniformity of both networks, these hydrogels are concurrently double network and homocomposite, which enables to adjust their unusual and potentially practically useful properties. The fabrication of such designer materials is enabled by the ability of the SDCs to control media viscoelasticity and composite stiffness. The SDCs bring about rapid physical solidification due to the attractive van der Waals interactions between their nanofibrillar coronas. This allows extrudability and injectability which are key prerequisites for numerous applications of hydrogels. The HHGs thus present a solution to the problem which makes molecular alginates and other common gel precursors poorly suitable for shaping by extrusion (such as in 3D printing and injection) because of a lack of initial yield stress which renders them fluid for some time after extrusion.

While the alginate–alginate system demonstrates the advantages of homocomposite formulations, the potential of SDCs as

viscoelastic modification and yield stress additives can be extended to many other systems. The SDCs are highly efficient as homocomposite fillers and their dispersion within other continuous polymer phases allows precise modulation of their viscoelasticity and composite stiffness at extremely low filler concentrations. The physical gelling of SDC as means of imparting controlled yield stress to a mixture that can be immediately 3D printed and subsequently builds up strength can also be applied in heterocomposite formulations. To illustrate how SDC reinforcing networks can be used in heterocomposite gel formulations, we constructed several examples, which are described in the Supplementary Information. The first example is a gel with liquid silicone matrix and polyvinyl alcohol SDCs that exhibits a yield stress at extremely dilute SDC concentrations and forms polydimethylsiloxane elastomer inks suitable for 3D printing by extrusion (Supplementary Fig. 8a). In addition, we combined the "homocomposite" approach presented herein with the well-studied alginate-polyacrylamide double network[1]. The resulting PAAm – SA CMH – SA SDC triple network gel showed increased tensile strength with the inclusion of the colloidal SDC network without increasing the total alginate concentration (Supplementary Fig. 8b).

The SA HHG examples unveiled here have numerous potential applications such as in biomedical 3D printing[32,63–65], soft robotics[3], food products with 3D printed edible structures[66], and

haptic materials with unique mechanical responses[67]. They could also enhance the design of green and sustainable materials made from biodegradable polymers and injectable medical hydrogels made of one pure component. Their most impactful feature is that the homocomposite colloidal-molecular double network makes possible independent tuning of chemical composition and material properties. This is enabled by the SDCs, which emerge as a class of soft matter capable of tailoring the properties of many liquid and solid composite materials.

## Methods

**Fabrication and characterization of SA SDCs**. Alginic acid sodium salt (SA, Sigma, 120–190 kDa) was dissolved in distilled water (Millipore) at 3 wt.%. Following dissolution, the mixture was stored at 4 °C. To fabricate SA SDCs, a colloidal high-shear mixer (IKA Magic Lab) was first filled with 500 mL of 13.6 mM calcium ion precipitation medium ($CaCl_2 \cdot 2H_2O$, Sigma-Aldrich). A volume of 20 mL of 3 wt.% SA solution was injected directly into the shear zone of the device at 20,000 rpm using a machined injection attachment, where the SDCs form nearly instantaneously via ionic precipitation and are left suspended in the precipitation medium. SDC suspensions were centrifuged for 3 minutes at $3.0 \times 10^3 \times g$, resulting in the formation of a concentrated SDC pellet. The compacted SDCs were then re-suspended to 8× their pellet volume in distilled water using a Vortex mixer. This process was repeated 5× to remove any remaining ions from the suspension. Following particle washing, the SDCs were centrifuged and diluted to their desired concentration. Particle morphologies were visualized using optical microscopy (BX-61 Olympus) and field emission scanning electron microscopy (FEI Verios 460 L scanning electron microscope). Images were processed using ImageJ to adjust brightness and contrast.

**Preparation of the molecularly crosslinked SA hydrogels**. Conventional CMHs were prepared by adding 0.1 wt.% $CaCO_3$ nanoparticles (NPs, 10–80 nm, American Elements®) and 28.14 mM solution of D-Gluconic acid δ-lactone (GDL, Sigma-Aldrich) to an aqueous SA solution[60]. Upon dissolution in water and addition to the alginate mixture, GDL slowly hydrolyzes, reducing the pH and gradually ionizing the $CaCO_3$ NPs to release $Ca^{2+}$ crosslinker ions[61]. This is preferred over the addition of instantly soluble Ca salt (like $CaCl_2$) since it allows more homogeneous SA gel formation.

**Preparation of alginate homocomposite hydrogels**. Alginate molecular/SDC gels were made by first preparing stock solutions of 3.0 wt.% SA in water. These stock SA solutions were mixed with 0.30 wt.% $CaCO_3$ nanoparticles, and 1.38 wt.% SA SDCs in water and stirred thoroughly. To induce homogenous gelation, an aliquot of freshly prepared 20 wt.% D-(+)-gluconic acid δ-lactone in water was added to the mixture at a ratio of 5/1 (GDL/$CaCO_3$ w/w) and homogenized using a Vortex device. All gels were given 2 h to complete crosslinking prior to characterization. The final concentrations of each constituent of HHGs are shown in Supplementary Table 1.

**Rheological and tensile characterization of suspensions and hydrogels**. The viscoelastic properties of the SDC suspensions and HHGs were evaluated using a rheometer (Discovery HR-2, TA Instruments) equipped with a sandblasted plate and plate geometry (40 mm diameter and 0.8 mm gap size) with the temperature maintained at 25 °C for all experiments. Small amplitude oscillatory frequency sweeps were performed within the linear viscoelastic regime. Amplitude sweeps were performed between 0.1 and 1000% strain at a frequency of 6.23 rad/s and time sweeps were performed at a frequency of 1 rad/s with 1% strain. The tensile properties of the gels were determined using a common testing machine (Instron 5943) with samples of 1.5 mm thickness, 44 mm width, 10 mm length, and a 5 cm $min^{-1}$ crosshead speed. Laser-cut acrylic molds were used to shape hydrogels during equilibration and laser-cut acrylic grips were used for tensile analysis to avoid directly gripping the hydrogel sample. A thin layer of cyanoacrylate adhesive (Gorilla Glue Gel) was used for chemically bonding the gel to the grips to ensure that slippage did not occur during extension (Supplementary Fig. 9).

**X-ray photoelectron spectroscopy measurements**. XPS measurements were performed using a SPECS FlexMod instrument with Mg kα excitation (1254 eV). A hemispherical PHOIBIS 150 analyzer was used with a takeoff angle normal to the surface, a 30° X-ray incidence angle, and a 60° angle from X-ray source to analyzer. Energy calibration was established by referencing to adventitious Carbon (C1s line at 285.0 eV binding energy) and the base pressure in the analysis chamber is in $10^{-10}$ mbar range.

## Data availability statement

All data available on request from the authors.

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

## Acknowledgements

This study was supported by a grant from US National Science Foundation, no. CMMI-1825476. L.C.H. and A. R. J. are supported by a grant from the National Science Foundation (NSF CBET-1804462). We thank Dr. Michael Dickey for the use of his instrumentation for mechanical characterization of hydrogel samples. This work was performed in part at the Analytical Instrumentation Facility (AIF) at North Carolina State University, which is supported by the State of North Carolina and the National Science Foundation (award number ECCS-2025064). The AIF is a member of the North Carolina Research Triangle Nanotechnology Network (RTNN), a site in the National Nanotechnology Coordinated Infrastructure (NNCI).

## Author contributions

A.W., S.R., S.S., and O.V. initiated this work with A.W. and S.R. contributing equally. Experiments and analysis were completed by A.W., A.J., L.H., and S.R. All schematics shown are original and produced by A.W. and S.R. This work was advised by O.V. and S.S., and L.H., A.W., S.R., A.J., L.H., and O.V. wrote the manuscript.

## Competing interests

The authors declare no competing interests.

## Additional information

**Peer review information** *Nature Communications* thanks the anonymous reviewers for their contribution, to the peer review of this work. Peer reviewer reports are available.

