## [Peer Review File · Nature Communications]

REVIEWER COMMENTS

Reviewer #1 (Remarks to the Author):

This article is a comprehensive report of the synthesis and characterization of homocomposite hydrogels, wherein the polymer chemistry is consistent throughout the material, but control over micro and nanostructure allows access to mechanical properties of alginate gels that are very different from the “traditional” molecular gel alone. The authors use alginate as the material, which is a very common ionically crosslinked polymer widely used by many scientists and engineers, meaning that this method and material would be of broad interest and application. The results are presented clearly and in high quality, and the paper was easy to follow. The characterization is thorough with appropriate statistics. Authors even go on to show preliminary results from 3D printing of the gels. I think this is a nice paper that is worthy of acceptance.

A few minor points and suggestions.

-Sometimes for concentration values, wt% is given, but other times just % is stated. Please be consistent and put wt% if indeed it is weight percent for all concentrations

-Figure 5b, the symbols in the key are all squares instead of different shapes like the data on the plot

-What sort of aspect ratios can be printed with the gels?

-Please put a scale bar on the movies, especially the 3D printing one.

-Fig 3d schematic could benefit from a key or zoom in on the calcium ion crosslinking regions to actually show how the ions interact with the alginate. I see pictures that look like the egg box model, but this is not addressed and someone not already familiar with the chemical interactions in calcium - alginate would not understand what is going on. (This may also be a relevant comment for the schematics of the gels elsewhere – there are no actual chemical structures anywhere to show the molecule- ion interactions)

Reviewer #2 (Remarks to the Author):

Williams et al. report on a new type of homocomposite gels where the filler and the matrix both are composed of the same material, alginate. The authors conduct a comprehensive study into the chemical and mechanical characterization of these remarkable materials. As an application example, the authors further demonstrate printing of these gels into scaffold materials. Overall, this is a very well written paper that supports the notion of colloidal materials synergistically reinforcing hydrogels. The conclusions, in particular as they relate to the mechanical properties, are clear and supported by the data provided in 5 composite figures. The departure into the printing experiments demonstrate broader technical applicability and clearly elevates the potential impact of this work. In my view, the paper should be published after the following, minor comments are addressed:

1) The gelation of alginate hydrogels should in principle be reversible. Can the authors show reversibility in the mechanical properties, or maybe even degradability? How will the colloidal reinforcement elements behave relative to the matrix?

2) The authors refer to the interactions between the particular colloidal filler and the matrix as “synergistically enforced”. What would happen if the authors used another filler material that is not compositionally identical, but similar. Let’s say cellulose. Would the alginate colloids really provide the type of benefits claimed in the manuscript? A head-to-head comparison of selected mechanical properties could address this concern.

- 3) The authors clearly demonstrate the importance of the Ca^{2+} concentration. The alginate Ca binding is reversible. Under ambient conditions and changing ion concentrations, will this have any impact on the stability of the scaffolds? Could this potentially limit their applicability?
- 4) It would potentially be interesting to further understand the structural properties of the homocomposite gels. Thus, a structural characterization of the homocomposite gels using SANS to obtain gel properties, such as mesh size, etc. would further improve the manuscript.

Reviewer #3 (Remarks to the Author):

This paper in detailed reports a new concept of synergetic effects from a turbulent shear-driven dendritic alginate structure in a continuous alginate hydrogel, which composes a new synthesized self-reinforced homocomposite hydrogels (HHGs). Also, this new kind of hydrogel can be extruded into mesh structures by 3D printing. The rheological tests, especially the storage modulus, well validate the foregoing hypothesis. However, the mechanical performance (e.g., shown in Figure 3(b) and (c)) didn't sufficiently support the claim of synergy. More complementary data and statistics analysis should be provided in order to convince the readers. Additionally, regarding the authors' assertion of 3D printing ability, it is not so plausible due to results (e.g., shown in Figure 5) where only a two-dimensional structure by the 3D printing technique was seen. Thus, I would support the publication of this work on Nature Communication if more solid evidence about the synergic effects (e.g., Figure 3) can be provided, along with more prominent three-dimensional architectures printed with this HHGs. Moreover, it would be more convincing if the authors could explain and demonstrate more on the applications of the HHGs.

Comment 1: The authors seem to use CaCO_3 nanoparticles and claim it over whole paper. What is the exact size of these nanoparticles? Does it play a key role in forming alginate crosslinked molecular hydrogels (CMHs)? What would happen if smaller of larger CaCO_3 , e.g. micron-scaled particles, was used?

Comment 2. Pg4 "The alginate SDCs used as reinforcement networks were fabricated using a turbulent shear-driven precipitation process for the manufacturing of polymer dendricolloids." What is the average size of these polymer dendricolloids? How consistent is the size of these polymer dendricolloids with the described production method? Does the size affect the modulus and yield stress of the HHGs?

Comment 3 : What does it mean "The SDCs are surrounded by coronas of nanofibers as thin as 10 nm in diameter." (Page 5, lines 6-7)?

Comment 4. On Page 7(lines 11-15), the author says: " To avoid syneresis, we worked further with CMHs of 10/1 SA/ CaCO_3 ratio. Interestingly, at this ratio the pure CMH and SDC gels of 1.0 wt.% had storage moduli of similar magnitude, even though these two systems are gelled by different physical mechanisms and are very dissimilar in their appearance (clear CMH gels vs. turbid SDCs suspensions)." How can we understand "at this ratio the pure CMH and SDC gels of 1.0 wt.%"? What does the 1.0 wt.% mean?

Comment 5: Obviously, Figure 3 is a key figure to support the core idea of synergistic effects from hierarchically branched soft dendritic colloids (SDCs). It is so important to verify the proposed

homocomposite hydrogel that has stronger mechanical properties than any one of its components, if the author can supply more solid evidence. However, Figure 3(b) and (c) did not provide abundant evidence to defend this idea, although Figure 3(a) did so. Could you supplement the data of 1.0% SDC in both Figure 3(b) (for tensile curve) and Figure 3 (c) (for Young's modulus, blue curve)? Also, could you provide the statistic error bar for Figure 3(b) and especially for complex modulus in (c)? Why are there so many kinks of tensile curves in Figure 3(b)? It should be clearly explained. Could you explain why there are two peaks in complex modulus in Figure 3 (c) or its mechanisms?

Comment 6. How does the homocomposite hydrogel in this work perform compared to other homocomposites for 3D printing reported in prior literature in terms of material yield stress and mechanical strength?

Comment 7. It seems that Figure 3(d) and Figure 4(d) are too similar, and there seems no new information in Figure 4(d). One of these schematics should be enough to depict the structural synergism of the double SDC and CMH networks.

Comment 8. The authors claimed that "However, they alone cannot be readily crosslinked produce a stiff elastic material (Supplementary Fig. 4)." on Page 11, lines 12-13. First, this sentence doesn't make sense. Pls check this sentence structure. Second, the nomination for hydrogel systems really is not consistent, and even confusing in Supplementary Fig. 4. It is difficult to identify the similarity and difference, for instance, "SA SDCs", "molecular alginate solutions with addition of CaCO₃ and GDL", "SA in H₂O + CaCO₃, GDL" and "SA SDC + CaCO₃, GDL" et.al.

Comment 9. The authors show only one layer of printed homocomposite hydrogels in Figure 5. But Figure 5(c) shows the diameter of the extruded filament is not that uniform, and the horizontal and vertical lines in the mesh in Figure 5(d) are not very straight. It should be that the late lines inevitably disturb and move the previously deposited lines especially at their joints. How about printing with overhanging parts? Is the material able to support overhanging parts or does sagging happen? How long does it take for the first layer to solidify? Would recommend the authors to explain more in this area because to be a functional soft material suitable for printing, parts with overhanging features eg. scaffolds. should also be taken into considerations. Moreover, it will be more convinced regarding good 3D printability of homocomposite hydrogels if the author can print and present a real three-dimensional structure rather than the current two-dimensional mesh. Because 3D printing of alginate-based hydrogels is not so novel that some papers have already achieved it, even though those works used different alginate constitutions.

Reviewer #1 (Remarks to the Author):

This article is a comprehensive report of the synthesis and characterization of homocomposite hydrogels, wherein the polymer chemistry is consistent throughout the material, but control over micro and nanostructure allows access to mechanical properties of alginate gels that are very different from the “traditional” molecular gel alone. The authors use alginate as the material, which is a very common ionically crosslinked polymer widely used by many scientists and engineers, meaning that this method and material would be of broad interest and application. The results are presented clearly and in high quality, and the paper was easy to follow. The characterization is thorough with appropriate statistics. Authors even go on to show preliminary results from 3D printing of the gels. I think this is a nice paper that is worthy of acceptance.

A few minor points and suggestions.

We thank Reviewer 1 for the positive feedback.

1.) Sometimes for concentration values, wt% is given, but other times just % is stated. Please be consistent and put wt% if indeed it is weight percent for all concentrations

The Reviewer's point is well taken as the clear formulation of the gels presented herein are essential to their properties. We have adjusted the notation to consistently use 'wt.%' throughout the manuscript.

2.) Figure 5b, the symbols in the key are all squares instead of different shapes like the data on the plot

We have addressed this point and updated Fig. 5.

3.) What sort of aspect ratios can be printed with the gels?

The ability to print vertically developed structures using homocomposite paste could indeed be important for many of their potential applications. While this paper is not focused on applied hydrogel 3D printing, we have now performed an additional cycle of experiments aimed at demonstrating vertically developed structures, while evaluating the Height/Width ratio of the shaped structures. The results are demonstrated in Figure 5c and discussed in the new text in page 14. These examples show the ability to print with a Height/Width ratio of 10:1, which is estimated to be the aspect ratio limit for HHG formulations containing 1.5 wt.% total alginate. The lateral size of these structures is consistent with the previously reported lower limit for resolution of extruded alginate bioinks [1]. An increase in the alginate concentration – either by increasing the concentration of SDCs, or CMH – could further increase the yield stress of the paste, along with the limit of the Height/Width ratio of the 3D printed structures. Finally, the

height of the features of the structures can be increased multiple times by staged printing where the underlying layers are allowed to solidify over time, as also commented in the text (p. 14).

4.) Please put a scale bar on the movies, especially the 3D printing one.

We have added scale bars to both Supplementary Videos to address this useful suggestion.

5.) Fig 3d schematic could benefit from a key or zoom in on the calcium ion crosslinking regions to actually show how the ions interact with the alginate. I see pictures that look like the egg box model, but this is not addressed and someone not already familiar with the chemical interactions in calcium - alginate would not understand what is going on. (This may also be a relevant comment for the schematics of the gels elsewhere – there are no actual chemical structures anywhere to show the molecule- ion interactions)

We appreciate the good suggestion. We have reorganized Fig. 3c and added an extra zoom-in image that includes the egg-box structure of alginate-calcium interaction to show both structures that comprise the molecular and colloidal networks.

Reviewer #2 (Remarks to the Author):

Williams et al. report on a new type of homocomposite gels where the filler and the matrix both are composed of the same material, alginate. The authors conduct a comprehensive study into the chemical and mechanical characterization of these remarkable materials. As an application example, the authors further demonstrate printing of these gels into scaffold materials. Overall, this is a very well written paper that supports the notion of colloidal materials synergistically reinforcing hydrogels. The conclusions, in particular as they relate to the mechanical properties, are clear and supported by the data provided in 5 composite figures. The departure into the printing experiments demonstrate broader technical applicability and clearly elevates the potential impact of this work. In my view, the paper should be published after the following, minor comments are addressed:

We thank the Reviewer for the positive and perceptive feedback.

1) The gelation of alginate hydrogels should in principle be reversible. Can the authors show reversibility in the mechanical properties, or maybe even degradability? How will the colloidal reinforcement elements behave relative to the matrix?

This is an interesting comment by the Reviewer, as the differences in Ca^{2+} binding fundamentally distinguish the two forms of alginate comprising our HHGs. Calcium-alginate hydrogels are known to be stable over a range of pHs and temperatures, but it is known the calcium ion crosslinks can be removed and the gelation can be reversed by strong chelating agents such as EDTA [2]. We performed a corresponding set of experiments on all of the systems studied – CMH (molecular gel), SDC gel and HHGs. We found that when both SDCs and a CMH are subjected to 50 mM EDTA solutions, the chelator sequesters Ca^{2+} and induces rapid loss of structure and dissolution of the gels back into the aqueous phase. Interestingly,

HHGs are much less prone to dissolution and remain as slowly dissolving gel mass for more than 24 hours. This increased stability is likely a result of the reduced mass transport of the chelator and ions in the HHG bulk. We briefly describe these findings in page 10 of the revised manuscript and added SI text and Supplemental Figure S5.

2) The authors refer to the interactions between the particular colloidal filler and the matrix as “synergistically enforced”. What would happen if the authors used another filler material that is not compositionally identical, but similar. Let’s say cellulose. Would the alginate colloids really provide the type of benefits claimed in the manuscript? A head-to-head comparison of selected mechanical properties could address this concern.

The application of the principles reported in the paper in novel heterocomposite hydrogels, (vs. the present unique “homocomposite” ones) is undoubtedly important, although also extremely broad field. Molecular hydrogels reinforced with fibers and nanocellulose have been reported earlier, and in our view could not be a good basis for physical comparison. The novel and important element of the HHGs that we report is not the molecular medium, but the dendricolloid reinforcing matrix, with its fractal distributed network. Thus, we addressed the above comment by exploring and reporting the properties of non-homocomposite gels reinforced with alginate SDCs. The data conclusively demonstrate that the ability to use SDCs in traditional composites is also of importance as the Reviewer has rightfully noted. We have added in the revised manuscript a large set of data on the properties of two heterocomposite systems of specific interest. The first one is a gel with liquid silicone matrix and polyvinyl alcohol SDCs that exhibits yield stress at extremely dilute SDC concentrations and forms polydimethylsiloxane elastomer inks suitable for 3D printing by extrusion (Supplementary Fig. 8a and related SI text). In addition, we combined the “homocomposite” approach presented herein with the well-studied alginate-polyacrylamide double network. We obtained and included data using SDCs in “triple network” PAAm – SA CMH – SA SDC hydrogels showing increased mechanical properties of this system with SDC addition and without increasing total alginate concentration (Supplementary Fig. 8b and corresponding SI text). These novel types of gels are described in detail in the SI briefly introduced in the text (p.16). Obviously, the SDC templates have the potential to become a key component of many new heterocomposite hydrogel systems, but this topic mostly lies outside the scope of the present paper.

3) The authors clearly demonstrate the importance of the Ca²⁺ concentration. The alginate Ca binding is reversible. Under ambient conditions and changing ion concentrations, will this have any impact on the stability of the scaffolds? Could this potentially limit their applicability?

The Calcium-Alginate molecular hydrogels are a well investigated system, which is known to be stable over a wide range of pHs and temperatures.[2] The deterioration of the gel integrity in the presence of a few strong chelators such as EDTA, discussed in the response to pt. 1 above is also well known. If the hydrogels are used while continuously immersed in flowing water, it is possible that ionic leaching and mechanical changes occur, in a slower analogy to the use of chelators mentioned above. However, this phenomenon was never encountered in the experiments and the HHGs were very stable over long times. By maintaining a constant 10/1 SA/CaCO₃ ratio in the CMH component of HHGs, we mitigate gel instability by either syneresis

or Ca-ion leaching. We include an additional text explaining this in page 7 of the revised manuscript.

4) It would potentially be interesting to further understand the structural properties of the homocomposite gels. Thus, a structural characterization of the homocomposite gels using SANS to obtain gel properties, such as mesh size, etc. would further improve the manuscript.

SANS is indeed a powerful method for characterizing near-molecular structure and interactions, which was used earlier in Velev's research as well. Our expectation is that SANS may in the future provide supplemental information for the HHG system that may elucidate the intricate cross-linked structure at the interface between the colloidal network and the molecular gel. However, the sensitivity at which the surface effects will be revealed by SANS has yet to be evaluated, as the vast majority of the system consists of bulk molecular gel and bulk SDC dense gel. At this stage we believe that the most important source of future information of the system will be confocal microscopy of the colloidal network, but we also plan to explore the feasibility of SANS in potential future studies.

Reviewer #3 (Remarks to the Author):

This paper in detailed reports a new concept of synergetic effects from a turbulent shear-driven dendritic alginate structure in a continuous alginate hydrogel, which composes a new synthesized self-reinforced homocomposite hydrogels (HHGs). Also, this new kind of hydrogel can be extruded into mesh structures by 3D printing. The rheological tests, especially the storage modulus, well validate the foregoing hypothesis. However, the mechanical performance (e.g., shown in Figure 3(b) and (c)) didn't sufficiently support the claim of synergy. More complementary data and statistics analysis should be provided in order to convince the readers. Additionally, regarding the authors' assertion of 3D printing ability, it is not so plausible due to results (e.g., shown in Figure 5) where only a two-dimensional structure by the 3D printing technique was seen. Thus, I would support the publication of this work on Nature Communication if more solid evidence about the synergic effects (e.g. Figure 3) can be provided, along with more prominent three-dimensional architectures printed with this HHGs. Moreover, it would be more convincing if the authors could explain and demonstrate more on the applications of the HHGs.

We thank Reviewer 3 for the detailed comments and critique.

Comment 1: The authors seem to use CaCO₃ nanoparticles and claim it over whole paper. What is the exact size of these nanoparticles? Does it play a key role in forming alginate crosslinked molecular hydrogels (CMHs)? What would happen if smaller or larger CaCO₃, e.g. micron-scaled particles, was used?

The CaCO₃ nanoparticles serve as a source of Ca²⁺ ions for crosslinking. We have used a composition that is well-described in the literature (Ref. 58). The size range of the commercial CaCO₃ used in our project (10-80 nm) is now included in the revised Methods Section. Larger

particles could affect the CMH gel quality, as they could increase the heterogeneity of the gels by producing dense inhomogeneous Ca^{2+} cross-linked areas about the localities of the CaCO_3 particles. However, this method of gelation is well described in the literature and we did not observe any such problems. Our CMH gels were very homogeneous and optically clear, thus we expect that other researchers would be able to easily reproduce these compositions.

Comment 2. Pg4 “The alginate SDCs used as reinforcement networks were fabricated using a turbulent shear-driven precipitation process for the manufacturing of polymer dendricolloids.” What is the average size of these polymer dendricolloids? How consistent is the size of these polymer dendricolloids with the described production method? Does the size affect the modulus and yield stress of the HHGs?

The overall size of these enormously branched fibrous clusters is a parameter of secondary importance as their major feature is the hierarchical branching. However, there is an overall consistency in macroscopic properties of the SDCs as we now describe in page 5 in the revised paper. We have also added a Supplementary Figure with microscope data on the Feret diameter of spin-coated SDCs and made a histogram of SDC sizes to show their general range (Figure S1b). We note that the most important morphological feature of the dendricolloids is explained in more detail below.

Comment 3 : What does it mean “The SDCs are surrounded by coronas of nanofibers as thin as 10 nm in diameter.” (Page 5, lines 6-7)?

The soft dendritic colloids used in this study present a new class of soft matter morphology, where the micrometer-sized backbones are spreading out and branching hierarchically to nanofiber size. The bundle of fibers constituting an individual SDC is on the order of a couple hundred microns, but the branched thin flexible fibers in their outer layer or “corona” can be as thin as 10 nm. As demonstrated and analyzed in our paper describing the discovery and properties of the SDCs [3], the high surface area and van der Waals “stickiness” of these nanofibers lead to several unique structure-formation capabilities, which are used in the present study. Thus, the nanofiber outer layer is behind the very strong adhesivity between the coronas and the force distribution along the hierarchical networks. The Reviewer’s question points out to the importance of explaining the above points better in the context of the present work, which is now done in the revised manuscript. We have expanded this section of the manuscript (page 5) and added data and corresponding information in Supplementary Fig. 1.

Comment 4. On Page 7(lines 11-15), the author says: “To avoid syneresis, we worked further with CMHs of 10/1 SA/CaCO3 ratio. Interestingly, at this ratio the pure CMH and SDC gels of 1.0 wt.% had storage moduli of similar magnitude, even though these two systems are gelled by different physical mechanisms and are very dissimilar in their appearance (clear CMH gels vs. turbid SDCs suspensions).”. How can we understand “at this ratio the pure CMH and SDC gels of 1.0 wt.%”? What does the 1.0 wt.% mean?

We have reworded this portion in the manuscript (p. 7) to clarify that the wt.% in this context refers to the total amount of alginate in the suspension or hydrogel and have kept this notation consistent throughout the revised paper.

Comment 5: Obviously, Figure 3 is a key figure to support the core idea of synergistic effects from hierarchically branched soft dendritic colloids (SDCs). It is so important to verify the proposed homocomposite hydrogel that has stronger mechanical properties than any one of its components, if the author can supply more solid evidence. However, Figure 3(b) and (c) did not provide abundant evidence to defend this idea, although Figure 3(a) did so. Could you supplement the data of 1.0% SDC in both Figure 3(b) (for tensile curve) and Figure 3 (c) (for Young's modulus, blue curve)? Also, could you provide the statistic error bar for Figure 3(b) and especially for complex modulus in (c)? Why are there so many kinks of tensile curves in Figure 3(b)? It should be clearly explained. Could you explain why there are two peaks in complex modulus in Figure 3 (c) or its mechanisms?

Figure 3 summarizes a large body of rheology data, and we have tried to improve the presentation following Reviewer's comments. We have replotted Fig. 3b by way of average stress-strain curves with added error bars that approximate better the physical curves. As mentioned, Fig. 3d was also re-drawn. Our results demonstrating a synergistic effect (Figure 3c) have been repeated independently with multiple systems and clearly demonstrate the presence of such an effect. However, we could not collect data for the tensile strength of the pure SDC gels, for which the Reviewer asks: The brittle gel of pure SDCs does not allow us to measure its tensile strength, as the pure interlocked dendrimers form a matter close to fragile solid than elastic gel (see also Supplementary Figures S2 and S6). This illustrates the unique and complex behavior of the pure SDCs. Finally, our previous repeated measurements, as well as a new big experimental cycle performed while the paper was in revision to answer the Reviewer, prove the consistent existence of the small complex modulus peak at a composition of 0.875% SDCs, 0.125% SA CMH. We hypothesize that this slight increase of stability reflects the transition to near solid-like behavior at increased concentration of SDCs (same that leads to the tensile fragility at 1%). Obviously, this is a subtle and complex secondary effect, which will be investigated in further detail in the future. We have introduced new text to explain all above findings in the revised manuscript (pp. 8 and 9).

Comment 6. How does the homocomposite hydrogel in this work perform compared to other homocomposites for 3D printing reported in prior literature in terms of material yield stress and mechanical strength?

To our knowledge, we are the first to report homocomposite hydrogel with molecular-colloidal networks. Our paper also establishes a pioneering record in 3D printing with such hydrogels, as we are not aware of any earlier reports of such 3D printed homocomposite hydrogels. The homocomposite pastes for 3D printing in the prior literature are based on materials with totally different composition and morphology – they are porous silicone rubber materials, made by binding cured silicone beads with liquid silicone capillary bridges [4]. These silicone rubber homocomposites for 3D printing were also discovered in our group and have been a subject of intensive interest in very different aspects. They are cited here as a broadly relevant materials

concept, but as these systems are structurally and chemically different (porous silicone vs. alginate hydrogel), that they cannot be compared directly to the HHGs in the present paper.

Comment 7. It seems that Figure 3(d) and Figure 4(d) are too similar, and there seems no new information in Figure 4(d). One of these schematics should be enough to depict the structural synergism of the double SDC and CMH networks.

The Reviewer has rightly noticed that Figure 3d does not adequately illustrate both networks constituting the HHG. We have added the chemical “egg-box” structure in the revised Figure 3 to show molecular structure of calcium-alginate binding. In the revised Figure 4d, we emphasize the temporal dependence of the HHG on these two networks and that, prior to initiation of CMH cross-linking by GDL addition, the colloidal SDC network is still present and endows the yield stress behavior that enables 3D printing.

Comment 8. The authors claimed that “However, they alone cannot be readily crosslinked produce a stiff elastic material (Supplementary Fig. 4).” on Page 11, lines 12-13. First, this sentence doesn’t make sense. Pls check this sentence structure. Second, the nomination for hydrogel systems really is not consistent, and even confusing in Supplementary Fig. 4. It is difficult to identify the similarity and difference, for instance, “SA SDCs”, “molecular alginate solutions with addition of CaCO₃ and GDL”, “SA in H₂O + CaCO₃, GDL” and “SA SDC + CaCO₃, GDL” et.al.

We have reworded the sentence in question (p. 12). We note that the former Figure S4 is presently Fig. S6. We agree with the Reviewer that there is need for consistent notation of the different alginate phases and have unified the acronyms. We have also updated accordingly the Supplementary Information - Figs. S6, S11, S12.

Comment 9. The authors show only one layer of printed homocomposite hydrogels in Figure 5. But Figure 5(c) shows the diameter of the extruded filament is not that uniform, and the horizontal and vertical lines in the mesh in Figure 5(d) are not very straight. It should be that the late lines inevitably disturb and move the previously deposited lines especially at their joints. How about printing with overhanging parts? Is the material able to support overhanging parts or does sagging happen? How long does it take for the first layer to solidify? Would recommend the authors to explain more in this area because to be a functional soft material suitable for printing, parts with overhanging features eg. scaffolds. should also be taken into considerations. Moreover, it will be more convinced regarding good 3D printability of homocomposite hydrogels if the author can print and present a real three-dimensional structure rather than the current two-dimensional mesh.

The Reviewer has accurately pointed out that the capabilities of HHG 3D printing are not fully explored in this report, whose main goal is to introduce a physically new type of gel. The 3D printing with this hydrogel composition illustrates its versatility and capability to control the HHG rheology. Nevertheless, our 3D printing results are generally impressive and quite precise. The grid-like structures in Fig. 5a are composed of sine curves that have a gradient amplitude.

Because of this design, images may give the wrong impression that the grid-like structure is uneven.

We have added new data regarding the ability to extend the 3D additive printing via vertical layering. As shown in the newly added images in Figure 5c and corresponding text in page 14, we can easily stack up to 10 layers to a height of 1.2 cm using a printing nozzle of 1.2 mm inner diameter. Shaping of structures of much higher height can be achieved easily after staged printing with time breaks allowing the better consolidation of the gel structures. Achieving of overhanging structures was possible but exhibits some sagging. This is a well-known issue with hydrogel printing to the extent that other methods utilize dissolvable supports [5]. Overhanging can be optimized with further optimization of the method, but our paper is focused on innovative materials and physical principles, not 3D printing optimization.

... Because 3D printing of alginate-based hydrogels is not so novel that some papers have already achieved it, even though those works used different alginate constitutions.

We respectfully point out that the Reviewer seems to be critiquing something that we don't claim, and which has little relevance to the fundamental novelty of this paper. We have never claimed that our paper is the first one to report 3D printing with alginate. Our major claim to novelty is the fundamental breakthrough of introducing a principally new colloidal-molecular double network class of hydrogels. The unique properties of this new class of soft matter enable new capabilities, one of which is demonstrated by the 3D printing results. Even so, we point out that the new 3D hydrogel printing method has some distinctive advantages over the previously reported methods for 3D printing with alginate, where the rapid gelation is achieved by alginate injection inside a bath of Ca^{2+} solution, a cumbersome method that is much less versatile and practical than our elegant rheology-based approach. Thus, the Reviewer's comment prompted us to comment in the paper the important element of being able to print pure alginate gel without precipitation bath or similar aids used previously (p. 15).

References:

- [1] P. Rastogi, B. Kandasubramanian, Review of alginate-based hydrogel bioprinting for application in tissue engineering, *Biofabrication*. 11 (2019). <https://doi.org/10.1088/1758-5090/ab331e>.
- [2] P. Gurikov, I. Smirnova, Non-Conventional Methods for Gelation of Alginate, *Gels*. 4 (2018) 14. <https://doi.org/10.3390/gels4010014>.
- [3] S. Roh, A.H. Williams, R.S. Bang, S.D. Stoyanov, O.D. Velev, Soft dendritic microparticles with unusual adhesion and structuring properties, *Nat. Mater.* 18 (2019) 1315–1320. <https://doi.org/10.1038/s41563-019-0508-z>.
- [4] S. Roh, D.P. Parekh, B. Bharti, S.D. Stoyanov, O.D. Velev, 3D Printing by Multiphase Silicone/Water Capillary Inks, *Adv. Mater.* 29 (2017) 1–7. <https://doi.org/10.1002/adma.201701554>.
- [5] C. Xu, W. Chai, Y. Huang, R.R. Markwald, Scaffold-free inkjet printing of three-dimensional zigzag cellular tubes, *Biotechnol. Bioeng.* 109 (2012) 3152–3160. <https://doi.org/10.1002/bit.24591>.

REVIEWERS' COMMENTS

Reviewer #1 (Remarks to the Author):

I have read the response to the reviewers and looked over the changes made to the manuscript. The authors have addressed my concerns. I recommend publication.

Reviewer #2 (Remarks to the Author):

the authors have further improved the manuscript and addressed all my concerns

Reviewer #3 (Remarks to the Author):

Authors revised the manuscript well according to reviewers' comments. Also, I particularly appreciate authors' effort for the last figure with updated 3D printed hydrogel products. Honestly, as an expert in 3D printing processes, I focused more on potential of this hydrogel. Even though as a new material, this hydrogel is impressive and novel, I still cannot see good applications of this material since there have been a lot of approaches to print alginate and its composite systems, showing good mechanical and biological enhancement for tissue scaffolds. However, as a research article, I fully agree the novelty of this work with in-depth scientific information. Thus, I am happy to support this work for publication in this prestigious journal.

Reviewer #3 (Remarks to the Author):

Authors revised the manuscript well according to reviewers' comments. Also, I particularly appreciate authors' effort for the last figure with updated 3D printed hydrogel products. Honestly, as an expert in 3D printing processes, I focused more on potential of this hydrogel. Even though as a new material, this hydrogel is impressive and novel, I still cannot see good applications of this material since there have been a lot of approaches to print alginate and its composite systems, showing good mechanical and biological enhancement for tissue scaffolds. However, as a research article, I fully agree the novelty of this work with in-depth scientific information. Thus, I am happy to support this work for publication in this prestigious journal.

Response:

We appreciate the Reviewer's positive evaluation and recommendation for publication of our paper. We describe a new type of printable and extrudable hydrogel material with several unique properties. While the practical application of this material has yet to be accomplished in future work, we point out that the homocomposite has distinct advantages that can hardly be matched by other hydrogel compositions. Being made from only one pure component, our hydrogel can be especially valuable in biomedical applications. Since the manuscript was submitted, we have received feedback that this pure single component material can also be highly desirable for injectable compositions. We have included a brief note to that in the concluding paragraph (which paragraph was otherwise decreased in size following Editorial advice).

We note that nearly all highlighted changes in this second revised version are in response to your editorial comments. All of your comments have been taken into account. We hope that you find that this production-ready revised article is suitable for publication in *Nature Communications*. Thank you again for working on our manuscript.

Yours sincerely,

Orlin D. Velev
S. Frank and Doris Culberson Distinguished Professor
Dept. of Chemical and Biomolecular Engineering
North Carolina State University